# ADVERSARIAL AUTOAUGMENT

**Xinyu Zhang**
Huawei
zhangxinyu10@huawei.com

**Qiang Wang**
Huawei
wangqiang168@huawei.com

**Jian Zhang**
Huawei
zhangjian157@huawei.com

**Zhao Zhong**
Huawei
zorro.zhongzhao@huawei.com

## ABSTRACT

Data augmentation (DA) has been widely utilized to improve generalization in training deep neural networks. Recently, human-designed data augmentation has been gradually replaced by automatically learned augmentation policy. Through finding the best policy in well-designed search space of data augmentation, AutoAugment (Cubuk et al., 2019) can significantly improve validation accuracy on image classification tasks. However, this approach is not computationally practical for large-scale problems. In this paper, we develop an adversarial method to arrive at a computationally-affordable solution called ***Adversarial AutoAugment***, which can simultaneously optimize target related object and augmentation policy search loss. The augmentation policy network attempts to increase the training loss of a target network through generating adversarial augmentation policies, while the target network can learn more robust features from harder examples to improve the generalization. In contrast to prior work, we reuse the computation in target network training for policy evaluation, and dispense with the retraining of the target network. Compared to AutoAugment, this leads to about $12\times$ reduction in computing cost and $11\times$ shortening in time overhead on ImageNet. We show experimental results of our approach on CIFAR-10/CIFAR-100, ImageNet, and demonstrate significant performance improvements over state-of-the-art. On CIFAR-10, we achieve a top-1 test error of *1.36%*, which is the currently best performing single model. On ImageNet, we achieve a leading performance of top-1 accuracy *79.40%* on ResNet-50 and *80.00%* on ResNet-50-D without extra data.

## 1 INTRODUCTION

Massive amount of data have promoted the great success of deep learning in academia and industry. The performance of deep neural networks (DNNs) would be improved substantially when more supervised data is available or better data augmentation method is adapted. Data augmentation such as rotation, flipping, cropping, *etc.*, is a powerful technique to increase the amount and diversity of data. Experiments show that the generalization of a neural network can be efficiently improved through manually designing data augmentation policies. However, this needs lots of knowledge of human expert, and sometimes shows the weak transferability across different tasks and datasets in practical applications. Inspired by neural architecture search (NAS)(Zoph & Le, 2016; Zoph et al., 2017; Zhong et al., 2018a;b; Guo et al., 2018), a reinforcement learning (RL) (Williams, 1992) method called AutoAugment is proposed by Cubuk et al. (2019), which can automatically learn the augmentation policy from data and provide an exciting performance improvement on image classification tasks. However, the computing cost is huge for training and evaluating thousands of sampled policies in the search process. Although proxy tasks, i.e., smaller models and reduced datasets, are taken to accelerate the searching process, tens of thousands of GPU-hours of consumption are still required. In addition, these data augmentation policies optimized on proxy tasks are not guaranteed to be optimal on the target task, and the fixed augmentation policy is also sub-optimal for the whole training process.

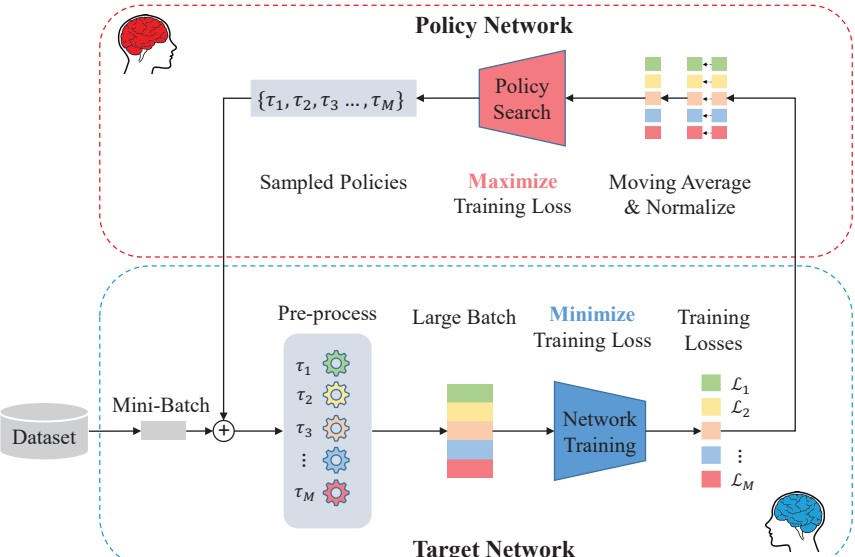

Figure 1: The overview of our proposed method. We formulate it as a Min-Max game. The data of each batch is augmented by multiple pre-processing components with sampled policies $\{\tau_1, \tau_2, \cdots, \tau_M\}$, respectively. Then, a target network is trained to minimize the loss of a large batch, which is formed by multiple augmented instances of the input batch. We extract the training losses of a target network corresponding to different augmentation policies as the reward signal. Finally, the augmentation policy network is trained with the guideline of the processed reward signal, and aims to maximize the training loss of the target network through generating adversarial policies.

In this paper, we propose an efficient data augmentation method to address the problems mentioned above, which can directly search the best augmentation policy on the full dataset during training a target network, as shown in Figure 1. We first organize the network training and augmentation policy search in an adversarial and online manner. The augmentation policy is dynamically changed along with the training state of the target network, rather than fixed throughout the whole training process like normal AutoAugment (Cubuk et al., 2019). Due to reusing the computation in policy evaluation and dispensing with the retraining of the target network, the computing cost and time overhead are extremely reduced. Then, the augmentation policy network is taken as an adversary to explore the weakness of the target network. We augment the data of each min-batch with various adversarial policies in parallel, rather than the same data augmentation taken in batch augmentation (BA) (Hoffer et al., 2019). Then, several augmented instances of each mini-batch are formed into a large batch for target network learning. As an indicator of the hardness of augmentation policies, the training losses of the target network are used to guide the policy network to generate more aggressive and efficient policies based on REINFORCE algorithm (Williams, 1992). Through adversarial learning, we can train the target network more efficiently and robustly.

The contributions can be summarized as follows:

- Our method can directly learn augmentation policies on target tasks, i.e., target networks and full datasets, with a quite low computing cost and time overhead. The direct policy search avoids the performance degradation caused by the policy transfer from proxy tasks to target tasks.

- We propose an adversarial framework to jointly optimize target network training and augmentation policy search. The harder samples augmented by adversarial policies are constantly fed into the target network to promote robust feature learning. Hence, the generalization of the target network can be significantly improved.

- The experiment results show that our proposed method outperforms previous augmentation methods. For instance, we achieve a top-1 test error of *1.36%* with PyramidNet+ShakeDrop (Yamada et al., 2018) on CIFAR-10, which is the state-of-the-art performance. On ImageNet, we improve the top-1 accuracy of ResNet-50 (He et al., 2016) from *76.3%* to *79.4%* without extra data, which is even *1.77%* better than AutoAugment (Cubuk et al., 2019).

## 2    RELATED WORK

Common data augmentation, which can generate extra samples by some label-preserved transformations, is usually used to increase the size of datasets and improve the generalization of networks, such as on MINST, CIFAR-10 and ImageNet (Krizhevsky et al., 2012; Wan et al., 2013; Szegedy et al., 2015). However, human-designed augmentation policies are specified for different datasets. For example, flipping, the widely used transformation on CIFAR-10/CIFAR-100 and ImageNet, is not suitable for MINST, which will destroy the property of original samples.

Hence, several works (Lemley et al., 2017; Cubuk et al., 2019; Lin et al., 2019; Ho et al., 2019) have attempted to automatically learn data augmentation policies. Lemley et al. (2017) propose a method called Smart Augmentation, which merges two or more samples of a class to improve the generalization of a target network. The result also indicates that an augmentation network can be learned when a target network is being training. Through well designing the search space of data augmentation policies, AutoAugment (Cubuk et al., 2019) takes a recurrent neural network (RNN) as a sample controller to find the best data augmentation policy for a selected dataset. To reduce the computing cost, the augmentation policy search is performed on proxy tasks. Population based augmentation (PBA) (Ho et al., 2019) replaces the fixed augmentation policy with a dynamic schedule of augmentation policy along with the training process, which is mostly related to our work. Inspired by population based training (PBT) (Jaderberg et al., 2017), the augmentation policy search problem in PBA is modeled as a process of hyperparameter schedule learning. However, the augmentation schedule learning is still performed on proxy tasks. The learned policy schedule should be manually adjusted when the training process of a target network is non-matched with proxy tasks.

Another related topic is Generative Adversarial Networks (GANs) (Goodfellow et al., 2014), which has recently attracted lots of research attention due to its fascinating performance, and also been used to enlarge datasets through directly synthesizing new images (Tran et al., 2017; Perez & Wang, 2017; Antoniou et al., 2017; Gurumurthy et al., 2017; Frid-Adar et al., 2018). Although we formulate our proposed method as a Min-Max game, there exists an obvious difference with traditional GANs. We want to find the best augmentation policy to perform image transformation along with the training process, rather than synthesize new images. Peng et al. (2018) also take such an idea to optimize the training process of a target network in human pose estimation.

## 3    METHOD

In this section, we present the implementation of ***Adversarial AutoAugment***. First, the motivation for the adversarial relation between network learning and augmentation policy is discussed. Then, we introduce the search space with the dynamic augmentation policy. Finally, the joint framework for network training and augmentation policy search is presented in detail.

### 3.1    MOTIVATIONS

Although some human-designed data augmentations have been used in the training of DNNs, such as randomly cropping and horizontally flipping on CIFAR-10/CIFAR-100 and ImageNet, limited randomness will make it very difficult to generate effective samples at the tail end of the training. To struggle with the problem, more randomness about image transformation is introduced into the search space of AutoAugment (Cubuk et al., 2019) (described in Section 3.2). However, the learned policy is fixed for the entire training process. All of possible instances of each example will be send to the target network repeatedly, which still results in an inevitable overfitting in a long-epoch training. This phenomenon indicates that the learned policy is not adaptive to the training process of a target network, especially found on proxy tasks. Hence, the dynamic and adversarial augmentation policy with the training process is considered as the crucial feature in our search space.

Another consideration is how to improve the efficiency of the policy search. In AutoAugment (Cubuk et al., 2019), to evaluate the performance of augmentation policies, a lot of child models should be trained from scratch nearly to convergence. The computation in training and evaluating the performance of different sampled policies can not be reused, which leads to huge waste of computation resources. In this paper, we propose a computing-efficient policy search framework through reusing prior computation in policy evaluation. Only one target network is used to evaluate the performance of different policies with the help of the training losses of corresponding augmented

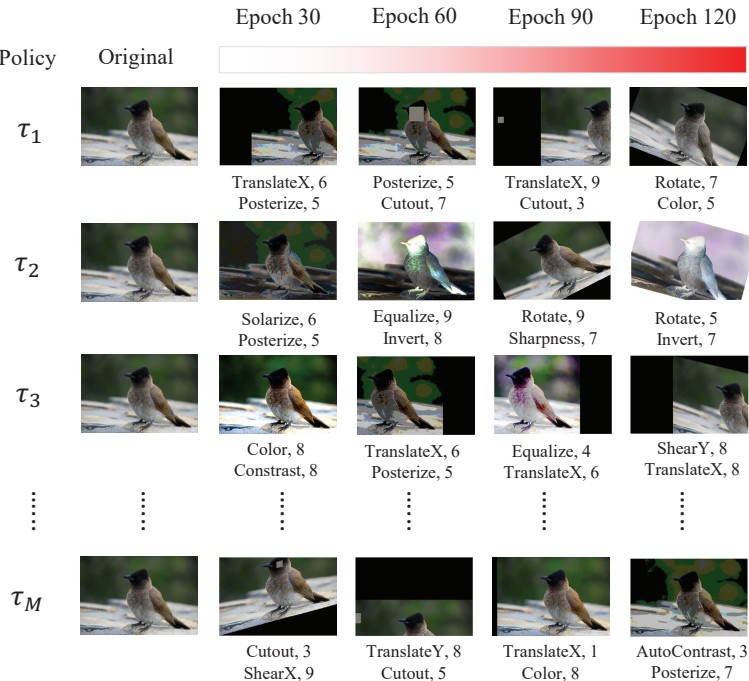

Figure 2: An example of dynamic augmentation policies learned with ResNet-50 on ImageNet. With the training process of the target network, harder augmentation policies are sampled to combat overfitting. Intuitively, more geometric transformations, such as TranslateX, ShearY and Rotate, are picked in our sampled policies, which is obviously different from AutoAugment (Cubuk et al., 2019) concentrating on color-based transformations.

instances. The augmentation policy network is learned from the intermediate state of the target network, which makes generated augmentation policies more aggressive and adaptive. On the contrary, to combat harder examples augmented by adversarial policies, the target network has to learn more robust features, which makes the training more efficiently.

## 3.2 SEARCH SPACE

In this paper, the basic structure of the search space of AutoAugment (Cubuk et al., 2019) is reserved. An augmentation policy is defined as that it is composed by 5 sub-policies, each sub-policy contains two image operations to be applied orderly, each operation has two corresponding parameters, i.e., the probability and magnitude of the operation. Finally, the 5 best policies are concatenated to form a single policy with 25 sub-policies. For each image in a mini-batch, only one sub-policy will be randomly selected to be applied. To compare with AutoAugment (Cubuk et al., 2019) conveniently, we just slightly modify the search space with removing the probability of each operation. This is because that we think the stochasticity of an operation with a probability requires a certain epochs to take effect, which will detain the feedback of the intermediate state of the target network. There are totally 16 image operations in our search space, including ShearX/Y, TranslateX/Y, Rotate, AutoContrast, Invert, Equalize, Solarize, Posterize, Contrast, Color, Brightness, Sharpness, Cutout (Devries & Taylor, 2017) and Sample Pairing (Inoue, 2018). The range of the magnitude is also discretized uniformly into 10 values. *To guarantee the convergence during adversarial learning, the magnitude of all the operations are set in a moderate range.*[1] Besides, the randomness during the training process is introduced into our search space. Hence, the search space of the policy in each epoch has $|S| = (16 \times 10)^{10} \approx 1.1 \times 10^{22}$ possibilities. Considering the dynamic policy, the number of possible policies with the whole training process can be expressed as $|S|^{\#epochs}$. An example of dynamically learning the augmentation policy along with the training process is shown in Figure 2. We observe that the magnitude (an indication of difficulty) gradually increases with the training process.

---

[1]The more details about the parameter setting please refer to AutoAugment (Cubuk et al., 2019).

### 3.3 Adversarial Learning

In this section, the adversarial framework of jointly optimizing network training and augmentation policy search is presented in detail. We use the augmentation policy network $\mathcal{A}(\cdot, \boldsymbol{\theta})$ as an adversary, which attempts to increase the training loss of the target network $\mathcal{F}(\cdot, \boldsymbol{w})$ through adversarial learning. The target network is trained by a large batch formed by multiple augmented instances of each batch to promote invariant learning (Salazar et al., 2018), and the losses of different augmentation policies applied on the same data are used to train the augmentation policy network by RL algorithm.

Considering the target network $\mathcal{F}(\cdot, \boldsymbol{w})$ with a loss function $\mathcal{L}[\mathcal{F}(\boldsymbol{x}, \boldsymbol{w}), \boldsymbol{y}]$, where each example is transformed by some random data augmentation $o(\cdot)$, the learning process of the target network can be defined as the following minimization problem

$$\boldsymbol{w}^* = \arg\min_{\boldsymbol{w}} \mathbb{E}_{\boldsymbol{x} \sim \Omega} \mathcal{L}[\mathcal{F}(o(\boldsymbol{x}), \boldsymbol{w}), \boldsymbol{y}], \tag{1}$$

where $\Omega$ is the training set, $\boldsymbol{x}$ and $\boldsymbol{y}$ are the input image and the corresponding label, respectively. The problem is usually solved by vanilla SGD with a learning rate $\eta$ and batch size $N$, and the training procedure for each batch can be expressed as

$$\boldsymbol{w}_{t+1} = \boldsymbol{w}_t - \eta \frac{1}{N} \sum_{n=1}^{N} \nabla_{\boldsymbol{w}} \mathcal{L}[\mathcal{F}(o(x_n), \boldsymbol{w}, y_n]. \tag{2}$$

To improve the convergence performance of DNNs, more random and efficient data augmentation is performed under the help of the augmentation policy network. Hence, the minimization problem should be slightly modified as

$$\boldsymbol{w}^* = \arg\min_{\boldsymbol{w}} \mathbb{E}_{\boldsymbol{x} \sim \Omega} \mathbb{E}_{\tau \sim \mathcal{A}(\cdot, \boldsymbol{\theta})} \mathcal{L}[\mathcal{F}(\tau(\boldsymbol{x}), \boldsymbol{w}), \boldsymbol{y}], \tag{3}$$

where $\tau(\cdot)$ represents the augmentation policy generated by the network $\mathcal{A}(\cdot, \boldsymbol{\theta})$. Accordingly, the training rule can be rewritten as

$$\boldsymbol{w}_{t+1} = \boldsymbol{w}_t - \eta \frac{1}{M \cdot N} \sum_{m=1}^{M} \sum_{n=1}^{N} \nabla_{\boldsymbol{w}} \mathcal{L}[\mathcal{F}(\tau_m(x_n), \boldsymbol{w}), y_n], \tag{4}$$

where we introduce $M$ different instances of each input example augmented by adversarial policies $\{\tau_1, \tau_2, \cdots, \tau_M\}$. For convenience, we denote the training loss of a mini-batch corresponding to the augmentation policy $\tau_m$ as

$$\mathcal{L}_m = \frac{1}{N} \sum_{n=1}^{N} \mathcal{L}[\mathcal{F}(\tau_m(x_n), \boldsymbol{w}), y_n]. \tag{5}$$

Hence, we have an equivalent form of Equation 4

$$\boldsymbol{w}_{t+1} = \boldsymbol{w}_t - \eta \frac{1}{M} \sum_{m=1}^{M} \nabla_{\boldsymbol{w}} \mathcal{L}_m. \tag{6}$$

Note that the training procedure can be regarded as a larger $N \cdot M$ batch training or an average over $M$ instances of gradient computation without changing the learning rate, which will lead to a reduction of gradient variance and a faster convergence of the target network Hoffer et al. (2019). However, overfitting will also come. To overcome the problem, the augmentation policy network is designed to increase the training loss of the target network with harder augmentation policies. Therefore, we can mathematically express the object as the following maximization problem

$$\boldsymbol{\theta}^* = \arg\max_{\boldsymbol{\theta}} J(\boldsymbol{\theta}),$$
$$\text{where } J(\boldsymbol{\theta}) = \mathbb{E}_{\boldsymbol{x} \sim \Omega} \mathbb{E}_{\tau \sim \mathcal{A}(\cdot, \boldsymbol{\theta})} \mathcal{L}[\mathcal{F}(\tau(\boldsymbol{x}), \boldsymbol{w}), \boldsymbol{y}]. \tag{7}$$

Similar to AutoAugment (Cubuk et al., 2019), the augmentation policy network is also implemented as a RNN shown in Figure 3. At each time step of the RNN controller, the softmax layer will

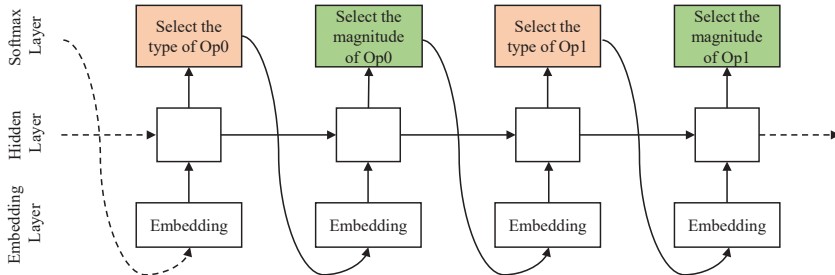

Figure 3: The basic architecture of the controller for generating a sub-policy, which consists of two operations with corresponding parameters, the type and magnitude of each operation. When a policy contains $Q$ sub-policies, the basic architecture will be repeated $Q$ times. Following the setting of AutoAugment (Cubuk et al., 2019), the number of sub-policies $Q$ is set to 5 in this paper.

predict an action corresponding to a discrete parameter of a sub-policy, and then an embedding of the predicted action will be fed into the next time step. In our experiments, the RNN controller will predict 20 discrete parameters to form a whole policy.

However, there has a severe problem in jointly optimizing target network training and augmentation policy search. This is because that non-differentiable augmentation operations break gradient flow from the target network $\mathcal{F}$ to the augmentation policy network $\mathcal{A}$ (Wang et al., 2017; Peng et al., 2018). As an alternative approach, REINFORCE algorithm (Williams, 1992) is applied to optimize the augmentation policy network as

$$
\begin{aligned}
\nabla_{\boldsymbol{\theta}} J(\boldsymbol{\theta}) &= \nabla_{\boldsymbol{\theta}} \underset{\boldsymbol{x} \sim \Omega}{\mathbb{E}} \underset{\tau \sim \mathcal{A}(\cdot, \boldsymbol{\theta})}{\mathbb{E}} \mathcal{L}[\mathcal{F}(\tau(\boldsymbol{x}), \boldsymbol{w}), \boldsymbol{y}] \\
&\approx \sum_m \mathcal{L}_m \nabla_{\boldsymbol{\theta}} p_m = \sum_m \mathcal{L}_m p_m \nabla_{\boldsymbol{\theta}} \log p_m \\
&= \underset{\tau \sim \mathcal{A}(\cdot, \boldsymbol{\theta})}{\mathbb{E}} \mathcal{L}_m \nabla_{\boldsymbol{\theta}} \log p_m \\
&\approx \frac{1}{M} \sum_{m=1}^{M} \mathcal{L}_m \nabla_{\boldsymbol{\theta}} \log p_m,
\end{aligned}
\tag{8}
$$

where $p_m$ represents the probability of the policy $\tau_m$. To reduce the variance of gradient $\nabla_{\boldsymbol{\theta}} J(\boldsymbol{\theta})$, we replace the training loss of a mini-batch $\mathcal{L}_m$ with $\widehat{\mathcal{L}}_m$ a moving average over a certain mini-batches[2], and then normalize it among $M$ instances as $\widetilde{\mathcal{L}}_m$. Hence, the training procedure of the augmentation policy network can be expressed as

$$
\begin{aligned}
\nabla_{\boldsymbol{\theta}} J(\boldsymbol{\theta}) &\approx \frac{1}{M} \sum_{m=1}^{M} \widetilde{\mathcal{L}}_m \nabla_{\boldsymbol{\theta}} \log p_m, \\
\boldsymbol{\theta}_{e+1} &= \boldsymbol{\theta}_e + \beta \frac{1}{M} \sum_{m=1}^{M} \widetilde{\mathcal{L}}_m \nabla_{\boldsymbol{\theta}} \log p_m,
\end{aligned}
\tag{9}
$$

The adversarial learning of target network training and augmentation policy search is summarized as Algorithm 1.

## 4 EXPERIMENTS AND ANALYSIS

In this section, we first reveal the details of experiment settings. Then, we evaluate our proposed method on CIFAR-10/CIFAR-100, ImageNet, and compare it with previous methods. Results in Figure 4 show our method achieves the state-of-the-art performance with higher computing and time efficiency[3].

---

[2]The length of the moving average is fixed to an epoch in our experiments.

[3]To clearly present the advantage of our proposed method, we normalize the performance of our method in the Figure 4, and the performance of AutoAugment is plotted accordingly.

---

**Algorithm 1** Joint Training of Target Network and Augmentation Policy Network

---

**Initialization:** target network $\mathcal{F}(\cdot, \boldsymbol{w})$, augmentation policy network $\mathcal{A}(\cdot, \boldsymbol{\theta})$
**Input:** input examples $\boldsymbol{x}$, corresponding labels $\boldsymbol{y}$

1: **for** $1 \leq e \leq epochs$ **do**
2:     Initialize $\widehat{\mathcal{L}}_m = 0, \forall m \in \{1, 2, \cdots, M\}$;
3:     Generate $M$ policies with the probabilities $\{p_1, p_2, \cdots, p_M\}$;
4:     **for** $1 \leq t \leq T$ **do**
5:         Augment each batch data with $M$ generated policies, respectively;
6:         Update $\boldsymbol{w}_{e,t+1}$ according to Equation 4;
7:         Update $\widehat{\mathcal{L}}_m$ through moving average, $\forall m \in \{1, 2, \cdots, M\}$;
8:     Collect $\{\widehat{\mathcal{L}}_1, \widehat{\mathcal{L}}_2, \cdots, \widehat{\mathcal{L}}_M\}$;
9:     Normalize $\widehat{\mathcal{L}}_m$ among $M$ instances as $\widetilde{\mathcal{L}}_m, \forall m \in \{1, 2, \cdots, M\}$;
10:    Update $\boldsymbol{\theta}_{e+1}$ via Equation 9;
11: Output $\boldsymbol{w}^*, \boldsymbol{\theta}^*$

---

## 4.1 EXPERIMENT SETTINGS

The RNN controller is implemented as a one-layer LSTM (Hochreiter & Schmidhuber, 1997). We set the hidden size to 100, and the embedding size to 32. We use Adam optimizer (Kingma & Ba, 2015) with a initial learning rate 0.00035 to train the controller. To avoid unexpected rapid convergence, an entropy penalty of a weight of 0.00001 is applied. All the reported results are the mean of five runs with different initializations.

## 4.2 EXPERIMENTS ON CIFAR-10 AND CIFAR-100

CIFAR-10 dataset (Krizhevsky & Hinton, 2009) has totally 60000 images. The training and test sets have 50000 and 10000 images, respectively. Each image in size of $32 \times 32$ belongs to one of 10 classes. We evaluate our proposed method with the following models: Wide-ResNet-28-10 (Zagoruyko & Komodakis, 2016), Shake-Shake (26 2x32d) (Gastaldi, 2017), Shake-Shake (26 2x96d) (Gastaldi, 2017), Shake-Shake (26 2x112d) (Gastaldi, 2017), PyramidNet+ShakeDrop (Han et al., 2017; Yamada et al., 2018). All the models are trained on the full training set.

**Training details:** The Baseline is trained with the standard data augmentation, namely, randomly cropping a part of $32 \times 32$ from the padded image and horizontally flipping it with a probability of 0.5. The Cutout (Devries & Taylor, 2017) randomly select a $16 \times 16$ patch of each image, and then set the pixels of the selected patch to zeros. For our method, the searched policy is applied in addition to standard data augmentation and Cutout. For each image in the training process, standard data augmentation, the searched policy and Cutout are applied in sequence. For Wide-ResNet-28-10, the step learning rate (LR) schedule is adopted. The cosine LR schedule is adopted for the other models. More details about model hyperparameters are supplied in A.1.

**Choice of $M$:** To choose the optimal $M$, we select Wide-ResNet-28-10 as a target network, and evaluate the performance of our proposed method verse different $M$, where $M \in \{2, 4, 8, 16, 32\}$. From Figure 5, we can observe that the test accuracy of the model improves rapidly with the increase of $M$ up to 8. The further increase of $M$ does not bring a significant improvement. Therefore, to balance the performance and the computing cost, $M$ is set to 8 in all the following experiments.

**CIFAR-10 results:** In Table 1, we report the test error of these models on CIFAR-10. For all of these models, our proposed method can achieve better performance compared to previous methods. We achieve $0.78\%$ and $0.68\%$ improvement on Wide-ResNet-28-10 compared to AutoAugment and PBA, respectively. We achieve a top-1 test error of $1.36\%$ with PyramidNet+ShakeDrop, which is $0.1\%$ better than the current state-of-the-art reported in Ho et al. (2019). As shown in Figure 6(a) and 6(b),we further visualize the probability distribution of the parameters of the augmentation policies learned with PyramidNet+ShakeDrop on CIFAR-10 over time. From Figure 6(a), we can find that the percentages of some operations, such as TranslateY, Rotate, Posterize, and SampleParing, gradually increase along with the training process. Meanwhile, more geometric transformations, such as TranslateX, TranslateY, and Rotate, are picked in the sampled augmentation policies, which is different from color-focused AutoAugment (Cubuk et al., 2019) on CIFAR-10. Figure 6(b) shows

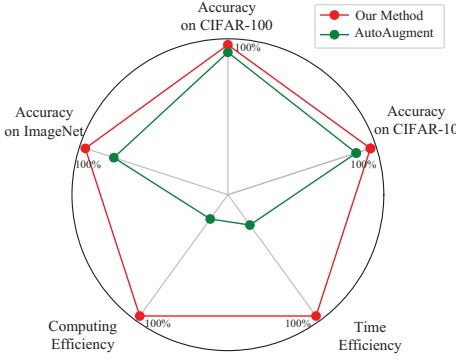

Figure 4: The Comparison of normalized performance between AutoAugment and our method. Please refer to the following tables for more details.

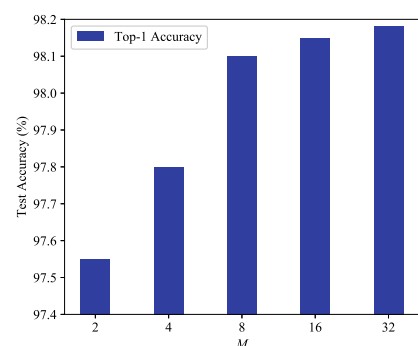

Figure 5: The Top-1 test accuracy of Wide-ResNet-28-10 on CIFAR-10 verse different $M$, where $M \in \{2, 4, 8, 16, 32\}$.

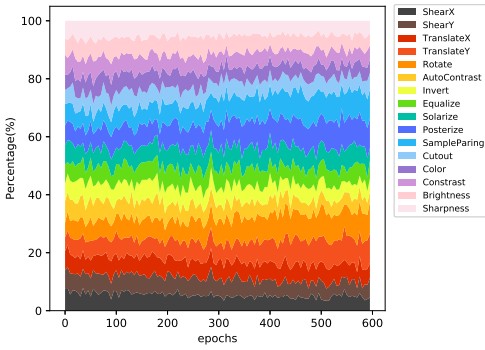

(a) Operations

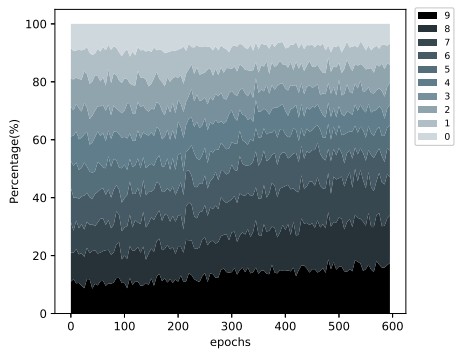

(b) Magnitudes

Figure 6: Probability distribution of the parameters in the learned augmentation policies on CIFAR-10 over time. The number in (b) represents the magnitude of one operation. Larger number stands for more dramatic image transformations. The probability distribution of each parameter is the mean of each five epochs.

that large magnitudes gain higher percentages during training. However, at the tail of training, low magnitudes remain considerable percentages. This indicates that our method does not simply learn the transformations with the extremes of the allowed magnitudes to spoil the target network.

**CIFAR-100 results:** We also evaluate our proposed method on CIFAR-100, as shown in Table 2. As we can observe from the table, we also achieve the state-of-the-art performance on this dataset.

Table 1: Top-1 test error (%) on CIFAR-10. We replicate the results of Baseline, Cutout and AutoAugment methods from Cubuk et al. (2019), and the results of PBA from Ho et al. (2019) in all of our experiments.

| Model | Baseline | Cutout | AutoAugment | PBA | Our Method |
|---|---|---|---|---|---|
| Wide-ResNet-28-10 | 3.87 | 3.08 | 2.68 | 2.58 | **1.90±0.15** |
| Shake-Shake (26 2x32d) | 3.55 | 3.02 | 2.47 | 2.54 | **2.36±0.10** |
| Shake-Shake (26 2x96d) | 2.86 | 2.56 | 1.99 | 2.03 | **1.85±0.12** |
| Shake-Shake (26 2x112d) | 2.82 | 2.57 | 1.89 | 2.03 | **1.78±0.05** |
| PyramidNet+ShakeDrop | 2.67 | 2.31 | 1.48 | 1.46 | **1.36±0.06** |

## 4.3 EXPERIMENTS ON IMAGENET

As a great challenge in image recognition, ImageNet dataset (Deng et al., 2009) has about 1.2 million training images and 50000 validation images with 1000 classes. In this section, we directly search the augmentation policy on the full training set and train ResNet-50 (He et al., 2016), ResNet-50-D (He et al., 2018) and ResNet-200 (He et al., 2016) from scratch.

Table 2: Top-1 test error (%) on CIFAR-100.

| Model | Baseline | Cutout | AutoAugment | PBA | Our Method |
|---|---|---|---|---|---|
| Wide-ResNet-28-10 | 18.80 | 18.41 | 17.09 | 16.73 | **15.49±0.18** |
| Shake-Shake (26 2x96d) | 17.05 | 16.00 | 14.28 | 15.31 | **14.10±0.15** |
| PyramidNet+ShakeDrop | 13.99 | 12.19 | 10.67 | 10.94 | **10.42±0.20** |

**Training details:** For the baseline augmentation, we randomly resize and crop each input image to a size of $224 \times 224$, and then horizontally flip it with a probability of $0.5$. For AutoAugment (Cubuk et al., 2019) and our method, the baseline augmentation and the augmentation policy are both used for each image. The cosine LR schedule is adopted in the training process. The model hyperparameters on ImageNet is also detailed in A.1.

**ImageNet results:** The performance of our proposed method on ImageNet is presented in Table 3. It can be observed that we achieve a top-1 accuracy $79.40\%$ on ResNet-50 without extra data. To the best of our knowledge, this is the highest top-1 accuracy for ResNet-50 learned on ImageNet. Besides, we only replace the ResNet-50 architecture with ResNet-50-D, and achieve a consistent improvement with a top-1 accuracy of $80.00\%$.

Table 3: Top-1 / Top-5 test error (%) on ImageNet. Note that the result of ResNet-50-D is achieved only through substituting the architecture.

| Model | Baseline | AutoAugment | PBA | Our Method |
|---|---|---|---|---|
| ResNet-50 | 23.69 / 6.92 | 22.37 / 6.18 | - | **20.60±0.15 / 5.53±0.05** |
| ResNet-50-D | 22.84 / 6.48 | - | - | **20.00±0.12 / 5.25±0.03** |
| ResNet-200 | 21.52 / 5.85 | 20.00 / 4.90 | - | **18.68±0.18 / 4.70±0.05** |

## 4.4 ABLATION STUDY

To check the effect of each component in our proposed method, we report the test error of ResNet-50 on ImageNet the following augmentation methods in Table 4.

- **Baseline**: Training regularly with the standard data augmentation and step LR schedule.
- **Fixed**: Augmenting all the instances of each batch with the standard data augmentation fixed throughout the entire training process.
- **Random**: Augmenting all the instances of each batch with randomly and dynamically generated policies.
- **Ours**: Augmenting all the instances of each batch with adversarial policies sampled by the policy network along with the training process.

From the table, we can find that Fixed can achieve $0.99\%$ error reduction compared to Baseline. This shows that a large-batch training with multiple augmented instances of each mini-batch can indeed improve the generalization of the model, which is consistent with the conclusion presented in Hoffer et al. (2019). In addition, the test error of Random is $1.02\%$ better than Fixed. This indicates that augmenting batch with randomly generated policies can reduce overfitting in a certain extent. Furthermore, our method achieves the best test error of $20.60\%$ through augmenting samples with adversarial policies. From the result, we can conclude that these policies generated by the policy network are more adaptive to the training process, and make the target network have to learn more robust features.

## 4.5 COMPUTING COST AND TIME OVERHEAD

**Computing Cost:** The computation in target network training is reused for policy evaluation. This makes the computing cost in policy search become negligible. Although there exists an increase of computing cost in target network training, the total computing cost in training one target network with augmentation policies is quite small compared to prior work.

**Time Overhead:** Since we just train one target network with a large batch distributedly and simultaneously, the time overhead of the large-batch training is equal to the regular training. Meanwhile, the joint optimization of target network training and augmentation policy search dispenses with the

Table 4: Top-1 test error (%) of ResNet-50 with different augmentation methods on ImageNet.

| Method | Aug. Policy | Enlarge Batch | LR Schedule | Test Error |
|---|---|---|---|---|
| Baseline | standard | $M = 1$ | step | 23.69 |
| Fixed | standard | $M = 8$ | cosine | 22.70 |
| Random | random | $M = 8$ | cosine | 21.68 |
| Ours | adversarial | $M = 8$ | cosine | **20.60** |

process of offline policy search and the retraining of a target network, which leads to a extreme time overhead reduction.

In Table 5, we take the training of ResNet-50 on ImageNet as an example to compare the computing cost and time overhead of our method and AutoAugment. From the table, we can find that our method is $12\times$ less computing cost and $11\times$ shorter time overhead than AutoAugment.

Table 5: The comparison of computing cost (GPU hours) and time overhead (days) in training ResNet-50 on ImageNet between AutoAugment and our method. The computing cost and time overhead are estimated on 64 NVIDIA Tesla V100s.

| Method | Computing Cost | | | Time Overhead | | |
|---|---|---|---|---|---|---|
| | Searching | Training | Total | Searching | Training | Total |
| AutoAugment | 15000 | 160 | 15160 | 10 | 1 | 11 |
| Our Method | $\sim$0 | 1280 | 1280 | $\sim$0 | 1 | 1 |

### 4.6 TRANSFERABILITY ACROSS DATASETS AND ARCHITECTURES

To further show the higher efficiency of our method, the transferability of the learned augmentation policies is evaluated in this section. We first take a snapshot of the adversarial training process of ResNet-50 on ImageNet, and then directly use the learned dynamic augmentation policies to regularly train the following models: Wide-ResNet-28-10 on CIFAR-10/100, ResNet-50-D on ImageNet and ResNet200 on ImageNet. Table 6 presents the experimental results of the transferability. From the table, we can find that a competitive performance can be still achieved through direct **policy transfer**. This indicates that the learned augmentation policies transfer well across datasets and architectures. However, compared to the proposed method, the policy transfer results in an obvious performance degradation, especially the transfer across datasets.

Table 6: Top-1 test error (%) of the transfer of the augmentation policies learned with ResNet-50 on ImageNet.

| Method | Dataset | AutoAugment | Our Method | Policy Transfer |
|---|---|---|---|---|
| Wide-ResNet-28-10 | CIFAR-10 | 2.68 | **1.90** | 2.45$\pm$0.13 |
| Wide-ResNet-28-10 | CIFAR-100 | 17.09 | **15.49** | 16.48$\pm$0.15 |
| ResNet-50-D | ImageNet | - | **20.00** | 20.20$\pm$0.05 |
| ResNet-200 | ImageNet | 20.00 | **18.68** | 19.05$\pm$0.10 |

## 5 CONCLUSION

In this paper, we introduce the idea of adversarial learning into automatic data augmentation. The policy network tries to combat the overfitting of the target network through generating adversarial policies with the training process. To oppose this, robust features are learned in the target network, which leads to a significant performance improvement. Meanwhile, the augmentation policy search is performed along with the training of a target network, and the computation in network training is reused for policy evaluation, which can extremely reduce the search cost and make our method more computing-efficient.

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

## A    APPENDIX

### A.1    HYPERPARAMETERS

We detail the model hyperparameters on CIFAR-10/CIFAR-100 and ImageNet in Table 7.

Table 7: Model hyperparameters on CIFAR-10/CIFAR-100 and ImageNet. LR represents learning rate, and WD represents weight decay. We do not specifically tune these hyperparameters, and all of these are consistent with previous works, expect for the number of epochs.

| Dataset | Model | Batch Size $(N \cdot M)$ | LR | WD | Epoch |
|---------|-------|--------------------------|-----|-----|-------|
| CIFAR-10 | Wide-ResNet-28-10 | $128 \cdot 8$ | 0.1 | 5e-4 | 200 |
| CIFAR-10 | Shake-Shake (26 2x32d) | $128 \cdot 8$ | 0.2 | 1e-4 | 600 |
| CIFAR-10 | Shake-Shake (26 2x96d) | $128 \cdot 8$ | 0.2 | 1e-4 | 600 |
| CIFAR-10 | Shake-Shake (26 2x112d) | $128 \cdot 8$ | 0.2 | 1e-4 | 600 |
| CIFAR-10 | PyramidNet+ShakeDrop | $128 \cdot 8$ | 0.1 | 1e-4 | 600 |
| CIFAR-100 | Wide-ResNet-28-10 | $128 \cdot 8$ | 0.1 | 5e-4 | 200 |
| CIFAR-100 | Shake-Shake (26 2x96d) | $128 \cdot 8$ | 0.1 | 5e-4 | 1200 |
| CIFAR-100 | PyramidNet+ShakeDrop | $128 \cdot 8$ | 0.5 | 1e-4 | 1200 |
| ImageNet | ResNet-50 | $2048 \cdot 8$ | 0.8 | 1e-4 | 120 |
| ImageNet | ResNet-50-D | $2048 \cdot 8$ | 0.8 | 1e-4 | 120 |
| ImageNet | ResNet-200 | $2048 \cdot 8$ | 0.8 | 1e-4 | 120 |

