# OpenReview forum: "Adversarial AutoAugment"
_ICLR.cc/2020/Conference — Accept (Poster)_

### Official Review · AnonReviewer2 · 2019-10-20
**Official Blind Review #2**

**Rating:** 6

**Review:**

This paper proposes a technique called Adversarial AutoAugment which dynamically learns good data augmentation policies during training. An adversarial approach is used: a target network tries to achieve good classification performance on a training set, while a policy network attempts to foil the target network by developing data augmentation policies that will produce images that are difficult to classify. To train the policy network, each mini-batch is augmented multiple times with different policies sampled from the policy network. Each augmented mini-batch is passed through the target network to produce a corresponding training loss, which is used as the training signal for the policy network. Experimental results are shown for CIFAR-10, CIFAR-100, and ImageNet datasets, where Adversarial AutoAugment outperforms competing methods (AutoAugment and Population Based Augmentation) on a variety of model architectures.

In its current state, I would tend towards rejecting this paper. The overall structure, the figures, and the experimental results are very nice, but there are two major issues that are holding it back. First, I am skeptical that the policy network is actually learning useful policies. Secondly, there are many grammatical errors in the paper which hamper readability. Upon reading the first paragraph of the paper, my initial impression was already quite negative, simply due to the number of grammatical errors. Fixing these would strengthen the paper considerably, and I would increase my score accordingly if properly addressed.

Primary Concerns:
1) One of my main concerns with this paper is this line here: "To guarantee the convergence during adversarial learning, the magnitude of all the operations are set in a moderate range". Since the policy network has no incentive to select transformations that the target network can still learn from, I assume that the ranges are required so that the policy network cannot choose to apply extreme transformations which always fool the target network, such as setting brightness to 0 to make the entire image black. How are acceptable ranges determined? If cross validation is required, then this becomes very much like the original hand-tuning of data augmentations that we wanted to avoid in the first place.

Additionally, I think it would be useful to to see a plot of the magnitude of each transformation versus training epoch, similar to Figure 4a in [1]. If the policy network simply learns to use the most extreme augmentations available in order to fool the target network, then this may indicate that the gain in performance is from tuning the magnitude ranges, and not from the policy network selecting good policies.

2) The paper could benefit greatly from some revision of the grammar. I would recommend either having a friend or colleague read it over, or even using an automated grammar checking program, such as Grammarly. For example, in the first paragraph alone there are several sentences that could be improved:
"Massive amount of data promotes the great success" -> "Massive amounts of data have promoted the great success"
"when more supervised data available" -> "when more supervised data is available"
"or better data augmentation method adapted" -> "or a better data augmentation method is adopted"
"which can automated learn" -> "which can automatically learn"
"there still requires tens of thousands of GPU-hours consumption." -> "tens of thousands of GPU-hours of computation are still required"


Things to improve the paper that did not impact the score:
3) In the first paragraph it is claimed that data augmentation policies have weak transferability across different tasks and datasets. I do not fully agree with this claim, since papers such as AutoAugment [2] have shown that learned policies are highly transferable to new datasets, and augmentation strategies such as CutMix have been shown to be highly effective for a variety of tasks.

4) No citation for the original GAN paper, despite multiple mentions of adversarial learning, and GANs themselves.

5) There is no computation time comparison with PBA, which is about 1000x faster than Autoaugment, and therefore roughly 100x faster than Adversarial AutoAugment.

6) CIFAR-10 results are not necessarily state-of-the-art. The 1.36% error rate claimed by the paper is surpassed by work in [3], which achieved 1.33% using a combination of AutoAugment and mixup.

7) Citation for the CIFAR-10 dataset incorrectly refers to the Adam optimizer paper [4].


References:
[1] Daniel Ho, Eric Liang, Ion Stoica, Pieter Abbeel, and Xi Chen. Population based augmentation: Efficient learning of augmentation policy schedules. ICML, 2019.

[2] Cubuk, Ekin D., Barret Zoph, Dandelion Mane, Vijay Vasudevan, and Quoc V. Le. "Autoaugment: Learning augmentation policies from data." CVPR (2019).

[3] Wistuba, Martin, Ambrish Rawat, and Tejaswini Pedapati. "A Survey on Neural Architecture Search." arXiv preprint arXiv:1905.01392 (2019).

[4] Diederik P. Kingma and Jimmy Ba. Adam: A method for stochastic optimization. ICLR, 2015

EDIT: The authors have addressed the majority of my concerns, and as such I have increased my score from a 3 to a 6.

**Experience Assessment:**

I have published one or two papers in this area.

**Review Assessment: Checking Correctness Of Derivations And Theory:**

N/A

**Review Assessment: Checking Correctness Of Experiments:**

I carefully checked the experiments.

**Review Assessment: Thoroughness In Paper Reading:**

I read the paper thoroughly.

---

> ### Author Response · Authors · 2019-11-11
> **Response to Review #2**
>
> We want to express our deep gratitude to for the constructive suggestions and positive comments on the novelty, motivation, and performance. We will explain your concerns point by point and hope you find our new revision satisfactory.
>
> >>> Response to “Whether the policy network is actually learning useful policies?”:
> 1)	In this paper, the generalization of the target network is improved by finding the best data augmentation policy to perform label-preserved image transformation. Although adversarial augmentation policies are generated by the policy network, the extreme and unmeaning image transformation that will almost destroy all the image information, such as making the entire image black or white, should not be included in the search space. This can be regarded as the guideline to determine the acceptable range of the magnitude of all the operations. How to handle extreme image transformations will be considered in our future works.
> 2)	We think that the ablation study has shown the effectiveness of adversarial augmentation policies. Through training the target network with adversarial augmentation policies, our method achieves the best test error, which indicates that the policy network can generate adversarial policies which are more adaptive to the training process, rather than apply extreme transformations which always fool the target network.
>
> >>> Response to “grammatical errors and typos”:
> Sincerely apologize for these mistakes. We have corrected these grammatical errors and typos, and revised the paper accordingly. Following your kind advice, we have also invited some native speakers to proofread the paper carefully. Wish you find our new revision satisfactory. Thanks again for pointing out our mistakes.
>
> >>> Response to “claim about weak transferability of human-designed augmentation policies”:
> We are very sorry for the misunderstanding. We just want to claim that human-designed augmentation policies sometimes show the weak transferability across different datasets, which is also presented in [1]. We have modified it in the new version.
>
> >>> Response to “No citation for the original GAN paper”:
> Thank you for the reference, which has been included in the new version.
>
> >>> Response to “No computation time comparison with PBA”:
> Because the experiment analysis on ImageNet dataset is not provided in PBA, we don’t compare computation time with it in Table 5. Although PBA is 1000x faster than AutoAugment in term of the searching cost, the cost of target network training is still non-trivial. Besides, we think that the comparison of computation time on large-scale tasks is more meaningful. Hence, we compare the total computing cost and time overhead in training ResNet-50 on ImageNet dataset between AutoAugment and our method.
>
> >>> Response to “CIFAR-10 results are not necessarily state-of-the-art”:
> We are very sorry for the omission. Due to the trivial difference, we think that the Top1 error of 1.36% on CIFAR-10 is comparable to the SOTA results in [2]. Meanwhile, we demonstrate that our overall performance is largely robust (insensitive) to the random factor due to the small variances on five runs in the revised version.
>
> >>> Response to “citation error”:
> Apologize for the typo. We have fixed it in the revised version.
>
> [1] Cubuk, Ekin D., Barret Zoph, Dandelion Mane, Vijay Vasudevan, and Quoc V. Le. "Autoaugment: Learning augmentation policies from data." CVPR (2019).
>
> [2] Wistuba, Martin, Ambrish Rawat, and Tejaswini Pedapati. "A Survey on Neural Architecture Search." arXiv preprint arXiv:1905.01392 (2019).

---

> > ### Comment · AnonReviewer2 · 2019-11-13
> > **Response to Authors**
> >
> > Thank you for taking the time to address my concerns. Please see some comments below.
> >
> > >>> Response to “Whether the policy network is actually learning useful policies?”:
> > The main reason I am skeptical about this goes back to the original AutoAugment [1] paper. In that paper they constrain the range of magnitudes for some transformation such that they do not produce images that might be out of the distribution of normal training images. This is a reasonable precaution to take. However, it turns out that those ranges were very important, since it was later shown that one did not even need to learn useful policies; randomly selecting some transformations from within the restricted range of magnitudes was good enough [2]. In essence, the very expensive policy search computation can be skipped if the pool of available transforms is restricted to a useful range, in which case we can revert to simple random sampling of augmentations.
> >
> > My main concern then is that the policy network in Adversarial AutoAugment is simply learning to apply transformations that are towards the extremes of the allowed ranges, since extreme transforms would be the most likely to fool the classifier. If we can achieve similar performance gains to Adversarial AutoAugment by simply restricting the augmentation pool to more extreme transformations, then we can again skip the computationally expensive search.
> >
> > If you can show that Adversarial AutoAugment is not simply learning to apply transformations that are on the extreme ranges of the allowed magnitudes, I would feel better about accepting this paper.
> >
> > [1] Cubuk, Ekin D., Barret Zoph, Dandelion Mane, Vijay Vasudevan, and Quoc V. Le. "Autoaugment: Learning augmentation policies from data." CVPR (2019).
> >
> > [2] Cubuk, Ekin D., Barret Zoph, Jonathon Shlens, and Quoc V. Le. "RandAugment: Practical data augmentation with no separate search." arXiv preprint arXiv:1909.13719 (2019).

---

> > > ### Author Response · Authors · 2019-11-14
> > > **Response to Review #2**
> > >
> > > Thank you for your valuable comments and suggestions. Hope your concerns can be well addressed in the new revision.
> > >
> > > >>> Response to “Whether the policy network is actually learning useful policies?”:
> > > We have visualized the probability distribution of the parameters in the learned augmentation policies on CIFAR-10 over training epochs in the new revision, and also added the analysis as follows:
> > >
> > > “As shown in Figure 6(a) and 6(b), we further visualize the probability distribution of the parameters of the augmentation policies learned with PyramidNet+ShakeDrop on CIFAR-10 over time. …. This indicates that our method does not simply learn the transformations with the extremes of the allowed magnitudes to spoil the target network.”
> > >
> > > The revision above indicates that the policy network does learn useful policies along with the training process rather than simply learn the policies with the extremes of the allowed magnitudes to spoil the target network.

---

> > > > ### Comment · AnonReviewer2 · 2019-11-14
> > > > **Response to Authors**
> > > >
> > > > Thank you for adding the additional plots, they are exactly what I was looking for.
> > > >
> > > > I am surprised that this method works so well without the policy network collapsing to just predicting extreme values. Although it does seem to be slowly converging towards using more extreme augmentations as training progresses. This leads me to a few more questions:
> > > >
> > > > 1. Would the policy network collapse to extreme values if the learning rate is too large/entropy penalty value too small, or if the number of epochs is increased?
> > > >
> > > > 2. How sensitive is the approach to the values of the controller's learning rate and the entropy penalty value? Can the same values used in the paper be used for any new dataset, or would they need to be re-tuned? Needing to tune these hyperparameters for each new dataset would be quite computationally expensive, and might negate any gains in efficiency over AutoAugment.

---

> > > > > ### Author Response · Authors · 2019-11-15
> > > > > **Response to Review #2**
> > > > >
> > > > > Thanks for your positive comments on our work.
> > > > >
> > > > > >>> Response to Q1:
> > > > > From our point of view, extreme learning rate or entropy penalty values may cause rapid convergence, which is unexpected. However, we do not think that equals to the collapsing results you worried about. We didn’t make specific experiments on those extreme hyperparameters, but we would like to make a deduction based on our understanding of the work:
> > > > > First, all magnitudes are set in a moderate range as mentioned before, which protect the network from collapsing during training even if they are picked at boundary.
> > > > > Then, with the training processes, loss of target network decreases gradually, which weaken the reward signal while converging policy network. Hence, extreme values are avoided during training.
> > > > >
> > > > > In Figure 6, the probability distribution of the parameters is visualized over 600 epochs (Appendix A1). We think this long-epoch training has made the policy network steadily converged, which can be observed from Figure 6(b). This indicates that the policy network may not collapse to extreme values given more epochs.
> > > > >
> > > > > >>> Response to Q2:
> > > > > We think the sensitivity analysis of the hyperparameters is benefit to consolidate our work, which would be considered in our future work due to the space and time limits.
> > > > > The setting of the hyperparameters is almost the same as AutoAugment. The extensive results have shown that it also works well on different datasets in our method, including CIFAR-10/100 and ImageNet. Hence, we have reason to believe that it probably could be used for other similar datasets. However, without exhaustive experiments, it is very difficult for us to determine whether it is suitable for any new dataset. If it is unfortunate that the hyperparameters are needed to be re-tuned for new datasets, we think AutoAugment perhaps will face the same problem during the policy search as well. According to the efficiency comparison between AutoAugment and our method in Table 5, we believe that the tuning of the hyperparameters for new datasets would not negate the gain in efficiency over AutoAugment.
> > > > >
> > > > > Hope our response answers your questions.

---

> > > > > > ### Comment · AnonReviewer2 · 2019-11-15
> > > > > > **Response to Authors**
> > > > > >
> > > > > > Thank you for your rebuttal and insightful comments. I think my main concerns have been addressed, and as such I will adjust my score accordingly.
> > > > > >
> > > > > > I'm not sure if there is time left to respond in the rebuttal period, but I was wondering if the learned augmentation policies could be distilled into a simple script such as done for AutoAugment in [1]? Since training the policy network is still quite computationally expensive and likely out of reach for the majority of practitioners, a data augmentation script with pretrained policies would likely be very well received.
> > > > > >
> > > > > > [1] https://github.com/DeepVoltaire/AutoAugment

---

> > > > > > > ### Author Response · Authors · 2019-11-15
> > > > > > > **Response to Review #2**
> > > > > > >
> > > > > > > Thank you very much. The helpful discussion improves the paper substantially. We will consider your suggestion.

---

### Official Review · AnonReviewer1 · 2019-10-27
**Official Blind Review #1**

**Rating:** 6

**Review:**

The authors propose a method for adversarial data augmentation which jointly trains the target network and the augmentation policy network. The authors claim that such learning set-up prevents overfitting and reduces computational cost with respect to competitors. Finally they provide extensive results and show that outperform the state-of-the-art in multiple datasets.

* The rationale behind the idea is properly introduced and justified
* The formulation is clear and sound
* The results show improvements over competitors in terms of accuracy and computational cost

* The contributions seem very incremental. Applying GANs for data augmentation is not new and regarding the policy search the authors strongly base their approach in Cubuk19. The main differentiator wrt the work of Cubuk is to jointly train both target and policy learner in a GAN setting, rather than learning the policies a priory with a fixed target network.

* I miss further discussion regarding the benefits of the GAN approach against pre-training policies. The overfitting argument seems pretty weak given the relatively marginal gains of accuracy.

* It would make the experiments more complete if the Top-1 and Top-5 results were provided, as well as running experiments on the same networks as the Cubuk19 paper.

* How coupled are the policies learned to the specific dataset after training?. The work by Cubuk19 shows transferability properties that could somehow diminish the gain achieved by this work regarding computational cost.


**Experience Assessment:**

I have published one or two papers in this area.

**Review Assessment: Checking Correctness Of Derivations And Theory:**

I assessed the sensibility of the derivations and theory.

**Review Assessment: Checking Correctness Of Experiments:**

I assessed the sensibility of the experiments.

**Review Assessment: Thoroughness In Paper Reading:**

I read the paper at least twice and used my best judgement in assessing the paper.

---

> ### Author Response · Authors · 2019-11-11
> **Response to Review #1**
>
> We want to express our deep gratitude to for the constructive suggestions and positive comments on the novelty, motivation, and performance.
>
> >>> Response to “incremental contributions”:
> We are very sorry for the misunderstanding. There are several reasons that strongly support for our substantial contributions.
> a)	Traditional GANs are used to enlarge datasets through directly synthesizing new images.  Although our method also plays in a GANs setting to jointly optimize target network and policy search, our goal is to find the best augmentation policy to perform label-preserved image transformations, rather than synthesize new images.
> b)	The exciting performance achieved by AutoAugment shows the potential benefits of automated data augmentation for training DNNs. However, the process of augmentation policy search in AutoAugment is very computationally expensive, and these augmentation policies learned on proxy tasks are not guaranteed to be optimal on target tasks. In this paper, we propose an adversarial framework to address these drawbacks. Through reusing the computing resource of target network training, the computing cost of policy search can be extremely degraded, as shown in Table 5. Directly learning policies on target tasks avoids the performance degradation caused by the transfer from proxy tasks to target tasks.
>
> >>> Response to “Benefits of the GAN approach against pre-training policies”:
> We are very sorry for confusing you. We think that the reported results have shown the benefits of the GAN approach against pre-training policies. Taking ResNet-50 trained on ImageNet dataset as an example, AutoAugment can achieve a drop of 1.32% in Top-1 error compared to baseline with the 15000 GPU*hours of consumption. However, our method achieves a drop of 3.09% in Top-1 error compared to baseline only with the (1280-160=1120) GPU*hours of additional consumption, which is even 1.77% better than AutoAugment. The transferability further shows that our method is more computing-efficient.
>
> >>> Response to “More complete results”:
> a)	The confidence interval with all the reported values and the Top-5 results on ImageNet have been provided in the new revision. We also list the table below:
> Table 3: Top-1 / Top-5 test error(%) on ImageNet
> +-------------+--------------+--------------+-----+--------------------------------+
> |    Model    |   Baseline   |  AutoAugment | PBA |           Our Method           |
> +-------------+--------------+--------------+-----+--------------------------------+
> |  ResNet-50  | 23.69 / 6.92 | 22.37 / 6.18 |  -  | 20.60$\pm$0.15 / 5.53$\pm$0.05 |
> +-------------+--------------+--------------+-----+--------------------------------+
> | ResNet-50-D | 22.84 / 6.48 |       -      |  -  | 20.00$\pm$0.12 / 5.25$\pm$0.03 |
> +-------------+--------------+--------------+-----+--------------------------------+
> |  ResNet-200 | 21.52 / 5.85 | 20.00 / 4.90 |  -  | 18.68$\pm$0.18 / 4.70$\pm$0.05 |
> +-------------+--------------+--------------+-----+--------------------------------+
>
> b)	We have tried our best to evaluate our method on the same networks as Cubuk19 except AmoebaNet.
> This is because our baseline training of AmoebaNet cannot achieve the reported accuracy in the original paper.
> The problem is also reported in [1].
>
> >>> Response to “couple the learned policies”:
> Generally, we don’t couple the learned policies again after target network training. This is because our method can directly perform the augmentation policy search along with target network training with a relatively low computing cost.However, to make our method more computing-efficient, we have shown the transferability of the learned policies between datasets and architectures in the new version.
>
> [1] Chen Lin, Minghao Guo, Chuming Li, Wei Wu, Dahua Lin, Wanli Ouyang, and Junjie Yan. Online
> hyper-parameter learning for auto-augmentation strategy. CoRR, abs/1905.07373, 2019.

---

### Official Review · AnonReviewer4 · 2019-10-29
**Official Blind Review #4**

**Rating:** 6

**Review:**

This paper describes a method to learn data augmentation policies using an adversarial loss. It builds on the AutoAugment method. In AutoAugment, an augmentation policy generator is trained by reinforcement learning. At each iteration, a classifier network is trained from scratch based on the current augmentation policy, and its validation accuracy is used as the reward signal. This is extremely costly because it requires to train a complete network for every training step of the policy generator. Instead, the current paper proposes to train the policy generator and the classifier simultaneously. The policy generator is trained adversarially to find augmentation policies that increase the loss of the classifier. This leads to a significant speedup compared to classic AutoAugment.

The presentation of the algorithm and the results is very clear. The proposed method yields improved performance compared to AutoAugment at ~1/10 of the computational cost, which is impressive. Although the paper only evaluates on two datasets (CIFAR and ImageNet), the idea is likely applicable very generally. I recommend this paper for publication, but have some comments that should be addressed:

Major comments:
- It would be good to evaluate how well the learned policies transfer between datasets and architectures. Adversarial AutoAugment still comes with a significant computational cost compared to hand-crafted augmentation, so transfer of policies would be useful. AutoAugment is transferable by design, so any competing algorithm should evaluate transferability.
- The authors state that all results are mean of 5 initializations, which is great. Please use these replicates to compute a measure of uncertainty (SEM or confidence interval) and state this with all values in the tables.

Minor comments:
- Overall, there are many grammatical errors and typos that sometimes require interpretation and reduce clarity. Please proof-read carefully.
- Abstract sentence “... can simultaneously optimizes…” has grammatical issues.
- Second sentence of introduction has grammatical issues.
- Third sentence of intro: “et al.” is used for persons, use “etc.” for things.
- Contribution section: “...our proposed method outperforms all previous augmentation method.” Please be careful with the breadth of your claims. You do not compare against *all* previous augmentation methods.
- Figure 4: Please add units and/or refer to Table 5 in the legend.


**Experience Assessment:**

I have read many papers in this area.

**Review Assessment: Checking Correctness Of Derivations And Theory:**

N/A

**Review Assessment: Checking Correctness Of Experiments:**

I carefully checked the experiments.

**Review Assessment: Thoroughness In Paper Reading:**

I read the paper at least twice and used my best judgement in assessing the paper.

---

> ### Author Response · Authors · 2019-11-11
> **Response to  Review #4**
>
> We want to express our deep gratitude to for the constructive suggestions and positive comments on the novelty, motivation, and performance.
>
> >>> Response to “evaluating transferability”:
> Thanks for your constructive advice. We have added the transferability analysis of the learned policies in the new version. Although the augmentation policies are dynamically changed in an adversarial manner, the learned policies can still transfer well to different datasets and architectures. Through directly applying the learned dynamic policies to various networks, the competitive performance is also achieved.
>
> >>> Response to “a measure of uncertainty”:
> Following your kind suggestion, we have stated the confidence interval with all the reported values in the new version. We also list the table below:
> Table 1: Top-1 test error(%) on CIFAR-10
> +-------------------------+----------+--------+-------------+------+---------------+
> |          Model          | Baseline | Cutout | AutoAugment |  PBA |   Our Method  |
> +-------------------------+----------+--------+-------------+------+---------------+
> |    Wide-ResNet-28-10    |   3.87   |  3.08  |     2.68    | 2.58 | 1.90$\pm$0.15 |
> +-------------------------+----------+--------+-------------+------+---------------+
> |  Shake-Shake (26 2x32d) |   3.55   |  3.02  |     2.47    | 2.54 | 2.36$\pm$0.10 |
> +-------------------------+----------+--------+-------------+------+---------------+
> |  Shake-Shake (26 2x96d) |   2.86   |  2.56  |     1.99    | 2.03 | 1.85$\pm$0.12 |
> +-------------------------+----------+--------+-------------+------+---------------+
> | Shake-Shake (26 2x112d) |   2.82   |  2.57  |     1.89    | 2.03 | 1.78$\pm$0.05 |
> +-------------------------+----------+--------+-------------+------+---------------+
> |   PyramidNet+ShakeDrop  |   2.67   |  2.31  |     1.48    | 1.46 | 1.36$\pm$0.06 |
> +-------------------------+----------+--------+-------------+------+---------------+
>
> Table 2: Top-1 test error(%) on CIFAR-100
> +------------------------+----------+--------+-------------+-------+----------------+
> |          Model         | Baseline | Cutout | AutoAugment |  PBA  |   Our Method   |
> +------------------------+----------+--------+-------------+-------+----------------+
> |    Wide-ResNet-28-10   |   18.80  |  18.41 |    17.09    | 16.73 | 15.49$\pm$0.18 |
> +------------------------+----------+--------+-------------+-------+----------------+
> | Shake-Shake (26 2x96d) |   17.05  |  16.00 |    14.28    | 15.31 | 14.10$\pm$0.15 |
> +------------------------+----------+--------+-------------+-------+----------------+
> |  PyramidNet+ShakeDrop  |   13.99  |  12.19 |    10.67    | 10.94 |  10.42$\pm$0.20 |
> +------------------------+----------+--------+-------------+-------+----------------+
>
> Table 3: Top-1 / Top-5 test error(%) on ImageNet
> +-------------+--------------+--------------+-----+--------------------------------+
> |    Model    |   Baseline   |  AutoAugment | PBA |           Our Method           |
> +-------------+--------------+--------------+-----+--------------------------------+
> |  ResNet-50  | 23.69 / 6.92 | 22.37 / 6.18 |  -  | 20.60$\pm$0.15 / 5.53$\pm$0.05 |
> +-------------+--------------+--------------+-----+--------------------------------+
> | ResNet-50-D | 22.84 / 6.48 |       -      |  -  | 20.00$\pm$0.12 / 5.25$\pm$0.03 |
> +-------------+--------------+--------------+-----+--------------------------------+
> |  ResNet-200 | 21.52 / 5.85 | 20.00 / 4.90 |  -  | 18.68$\pm$0.18 / 4.70$\pm$0.05 |
> +-------------+--------------+--------------+-----+--------------------------------+
> The small variances on 5 models demonstrate that our overall performance is largely robust (insensitive) to the random factor.
>
> >>> Response to “grammatical errors and typos”:
> Apologize for these errors and typos. We have corrected these grammatical errors and typos, and revised the paper accordingly.
>
> >>> Response to “the breadth of the claim”:
> Thanks for pointing out it. We have modified the claim in the new version.
>
> >>> Response to “Figure 4”:
> We have added the axis units on the figure.

---

> > ### Comment · AnonReviewer4 · 2019-11-14
> > **Response to authors**
> >
> > Thanks for making these changes, they address my comments.

---

### Decision · Program_Chairs · 2019-12-19

**Decision:**

Accept (Poster)

**Comment:**

This paper proposes a method to learn data augmentation policies using an adversarial loss. In contrast to AutoAugment where an augmentation policy generator is trained by RL (computationally expensive), the authors propose to train a policy generator and the target classifier simultaneously. This is done in an adversarial fashion by computing augmentation policies which increase the loss of the classifier. The authors show that this approach leads to roughly an order of magnitude improvement in computational cost over AutoAugment, while improving the test performance.
The reviewers agree that the presentation is clear and that the proposed method is sound, and that there is a significant practical benefit of using such a technique. As most of the concerns were addressed in the discussion phase, I will recommend acceptance of this paper. We ask the authors to update the manuscript to address the remaining (minor) concerns.